# Tacticity in chiral phononic crystals

A. Bergamini [1,3]*, M. Miniaci [1,3]*, T. Delpero[2], D. Tallarico [1], B. Van Damme[1], G. Hannema[1], I. Leibacher[2] & A. Zemp [1]

The study of vibrational properties in engineered periodic structures relies on the early intuitions of Haüy and Boscovich, who regarded crystals as ensembles of periodically arranged point masses interacting via attractive and repulsive forces. Contrary to electromagnetism, where mechanical properties do not couple to the wave propagation mechanism, in elasticity this paradigm inevitably leads to low stiffness and high-density materials. Recent works transcend the Haüy-Boscovich perception, proposing shaped atoms with finite size, which relaxes the link between their mass and inertia, to achieve unusual dynamic behavior at lower frequencies, leaving the stiffness unaltered. Here, we introduce the concept of tacticity in spin-spin-coupled chiral phononic crystals. This additional layer of architecture has a remarkable effect on their dispersive behavior and allows to successfully realize material variants with equal mass density and stiffness but radically different dynamic properties.

[1] Empa, Laboratory for Acoustics/Noise Control, Überlandstrasse 129, 8600 Dübendorf, Switzerland. [2] Empa, Laboratory for Structural Integrity of Energy Systems, Überlandstrasse 129, 8600 Dübendorf, Switzerland. [3]These authors contributed equally: A. Bergamini, M. Miniaci. *email: andrea.bergamini@empa.ch; marco.miniaci@empa.ch

The rational design of periodic structures has recently allowed to attain performances otherwise inaccessible by naturally available materials[1], such as being ultra-light and ultra-strong[2–5], hard to compress yet easy to deform[6] or simultaneously exhibiting negative stiffness, Poisson's ratio and/or mass density[7,8]. This offers unique potential for phonon manipulation[9] and wave isolation[10,11], including nonreciprocal[12–14] and scattering free[15,16] propagation, cloaking[17], and frequency bandgap nucleation[18].

Except for structures including locally resonant elements[19–22] or inertial amplification mechanisms[23], so far the tailoring of the dynamics of periodic structures has mainly relied on the Bragg scattering mechanism[24] and the exploration of their vibrational properties on the early intuitions of Haüy[25] and Boscovich[26], who regarded crystals as periodic ensembles of point masses that interact via attractive and repulsive forces (inter-atomic links). This allowed to conceptualize continuous as well as composite media as a series of discrete mass-spring systems[24,27] in analogy to the atoms and inter-atomic links of a crystal, predicting the essence of the wave propagation as a function of material density $\rho$, stiffness $\mathbb{C}$, and size of a representative unit cell $a$. However, the Haüy-Boscovich model necessarily considers the inter-atomic links as coaxial with the line connecting the atoms. Therefore, an atom can translate but there is no meaning in considering its rotation, leading to the concept of translational oscillator (TO), with a single degree of freedom (DOF) per space direction, the motion of which is described by the equation:

$$\ddot{u}m + uk = 0, \qquad (1)$$

being $m$ the mass of the oscillator, $k$ its axial stiffness and $u$ the displacement of the mass. Equation (1), applied to a mass-spring chain, implies that the vibrational properties of the TO and consequently the bandgap nucleation, are indissolubly linked to the stiffness and density of the material[24]. The constraint becomes more evident by normalizing the bandgap frequency as $\omega^* = f_{BG}a\sqrt{\frac{\rho}{\mathbb{C}}}$, being $f_{BG}$ the bottom edge frequency of the bandgap, $\rho$ the density and $\mathbb{C}$ the stiffness of the material. In the case of a 1D bi-atomic mass-spring chain we can show that $1/\pi \leq \omega^* \leq 2/\pi$ (see Supplementary Note 1), raising a fundamental theoretical limitation to the conception of structures with simultaneously high stiffness and low density but also small unit cell size and low-frequency bandgap, which has indeed been elusive so far. However, recent works[28–30] have proposed the use of chiral links to couple the longitudinal motion of finite size mass elements (atoms) to their rotation, referred to in what follows as spin, in analogy to the angular momentum carried by elementary particles in quantum mechanics. This inclusion of spin energy in the equation of motion allows to relax the bond between the aforementioned quantities determining $\omega^*$[28].

Here we show that, beyond the exploitation of inertial effects, the relative orientation of adjacent chiral centers, known from polymer science as tacticity, strongly affects the nature of the coupling between the spins of atoms in the chain. Our numerical and experimental investigations show that the syndiotactic phononic crystal arrangement nucleates low-frequency full bandgaps, while the isotactic variant exhibits wave modes in the same frequency range, responsible for a transmissive behavior. Nonetheless, the two variants are characterized by the same density and quasistatic stiffness. This is expected to have an impact on the understanding and design of chiral phononic crystals, making them a promising new class of materials for low frequency vibration isolation applications.

## Results

**Chirality.** The experimental confirmation of the above ideas, is obtained by preparing two samples, each comprising two unit cells arranged in the iso- and syndiotactic configuration, respectively, as shown in Fig. 1a, by means of an additive manufacturing process. The two structures essentially exhibit the same static stiffness in z-direction and have the same homogenized density (see Methods).

The transmissibility of the two systems is investigated by scanning laser Doppler vibrometry (SLDV) of the two samples (see Supplementary Fig. 1 for the description of the experimental set-up). The transmissibility is calculated as the ratio of the detected and imposed velocity along the principal axis at a scanning point and are presented in Fig. 1b, c. The clear drop in the transmission (Fig. 1c) can be seen as a direct observation of the numerically predicted bandgap. The experimental results are well supported by finite element numerical models, where the orthotropic material properties were obtained by matching the numerical model response of Fig. 1c to measured transmissibility data. Nominal data sheet material properties were used as an initial guess. Elastic constants derived from the transmission spectra are in excellent agreement with numerically simulated and experimental static tests (see Supplementary Fig. 2).

The introduction of chirality in phononic crystals transcends the Haüy-Boscovic crystal model by enriching the TO kinematics so as to weaken the link between the mass and the inertia of the atoms by means of non-centrosymmetric links coupling the translational motion of the atoms along one direction to their rotation about the same axis, in sharp contrast to ordinary Cauchy media (Fig. 2), which do not allow for chiral effects[31]. The concept of chirality has been long known in the literature and while it has been largely explored in the field of electromagnetic/optical metamaterials, its potential only recently emerged in elasticity[32–35]. Although substantial differences between the two domains (the absence of static chiral effects in optics and the mass density tensor additionally entering via the equation of motion in the elastic case) do not allow a direct transposition of concepts explored in one field to the other, a close mathematical analogy at the level of effective-medium description allows cross-fertilization inspiring new physics, such as negative refractive indices[36] or opening the path to the exploration of quasistatic properties of micropolar materials[37] and the dispersion relation of continua[38], among others.

The non-centrosymmetric architecture, made of elastic elements transferring axial and shear loads as well as bending moments, forces the atoms to rotate, clockwise (Fig. 2c) or counter-clockwise (Fig. 2d), under uni-axial tension, allowing for the conception of a coupled translational-rotational oscillator (TRO). The twist of the ligaments defines the direction of the atom rotation Φ with respect to its translation, leading to the definition of two variants of the chiral TRO, (+) and (−).

To unequivocally prove that the introduced spin allows to relax the restriction imposed by the Haüy-Boscovich model, we performed numerical forced frequency response analyses on the three oscillators reported in Fig. 2b–d, characterized by the same axial stiffness $k$ and mass $m$ (for further details, refer to Supplementary Notes 2 and 3). The amplitudes of the axial displacement as a function of the exciting frequency are reported in Fig. 2e for both the achiral TO (black line) and the two chiral TRO with enriched kinematics (superimposed dashed green and solid purple lines). While the harmonic response of the TO is determined by $\sqrt{\frac{k}{m}}$, the introduction of the spin into the unit cell kinematics leads to a considerable reduction of the first rigid atom mode (~30%), shifting the peak from 273 Hz in the case of the

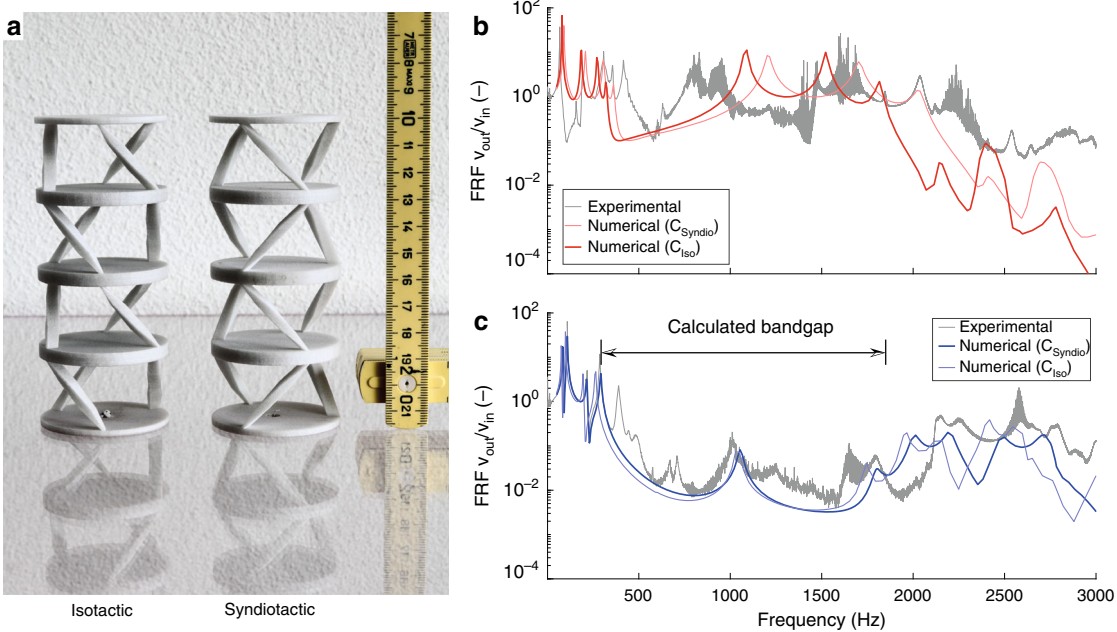

**Fig. 1** Measured and calculated transmission spectra. **a** Photograph of the investigated chiral phononic crystals. Isotactic sample on the left, syndiotactic on the right, with a cm scale, for reference. **b** Calculated (red) and measured (gray) transmission spectra of the isotactic crystal. The transmittance of the isotactic crystal falls below $10^{-1}$ at frequencies above 2200 Hz (i.e., above the O2 mode). This is due to the band structure of the isotactic crystal (see Fig. 4c) that presents few flat modes with small gaps in between them. **c** Calculated (blue) and measured (gray) transmission spectra of the syndiotactic crystal. The clear drop in the transmission can be seen as a direct observation of the numerically predicted bandgaps. It should be noted that a very simple damping model (proportional to the stiffness matrix) has been used for the finite element calculation of the transmission curves. Consequently, an attenuation proportional to the frequency is expected. The lines in strong color (dark red, dark blue) refer to the frequency response calculated based on the fit from the respective crystal, while the light color indicates the curve calculated using the properties obtained from the other crystal. While the position of features moves slightly, the character of the curves does not change

ordinary TO to 196 Hz in the case of a TRO. Furthermore, the reconstruction of the mode shapes clearly confirms the coupling mechanism between torsional and translational oscillation of the shaped atom, as shown in Fig. 2b–d and detailed in Supplementary Notes 2 and 3.

The enriched kinematics makes our work distinct from previous studies, as the frequency shift is solely due to the spin introduction and not to any stiffness or mass changes of the unit cell[19,39], and is described in the following section.

**Coupling of longitudinal and torsional motion.** The kinematics and statics of the coupling mechanism between translational and rotary motion of the inertia elements is detailed in Fig. 3. Here, for a matter of better understanding, the two main functions of the links between atoms are separately represented by different colors: (i) the blue spring elements, with stiffness constant $k$, provide a restoring force upon changes of the interatomic distance from the equilibrium condition (the analogue of the elastic elements in Brillouin's mass-spring chain), (ii) the black links, represent ideally stiff struts connected to the disks via frictionless pivots, imposing kinematic constraints to the motion of the gray disks by coupling the linear and rotational degrees of freedom, and (iii) the orange torsional springs provide restoring moment upon rotation of the atoms.

At the equilibrium condition, the struts connected at a distance $r$ from the axis of the unit cell and laying in a plane tangential to a circle, form an angle $\psi$ to the surface of the disks. The number of struts determines the order of the $n$-fold symmetry axis ($n = 3$, in the case at hand). For small displacements $u$, the rotation $\Phi$ of the

middle disk is given by:

$$\Phi r = \frac{u}{\tan(\psi)} \qquad (2)$$

The restoring force and moment originating from the elements with stiffness $k$ and $k_{ts}$ in longitudinal and rotary direction are:

$$F_k = k \cdot u, \qquad (3)$$

$$M_{ts} = k_{ts} \cdot \Phi, \qquad (4)$$

respectively. From the situation shown in Fig. 3 and the consideration of the kinematic constraints (2), we can write the equation of motion for the phononic crystal as follows:

$$\ddot{u}\left(m + \frac{\Theta}{\tan^2(\psi)r^2}\right) + u\left(k + \frac{k_{ts}}{\tan^2(\psi)r^2}\right) = 0. \qquad (5)$$

where $m$ is the mass of the atoms, $\Theta = \frac{1}{2}mr^2$, assuming that the struts are connected to the external circumference of the disk of radius $r$, $\psi$ is the angle of the struts between two adjacent disks, $k$ is the axial stiffness, and $k_{ts}$ is the torsional stiffness of the structure.

In spite of the fact that part of the energy is coupled into a rotary oscillation, the system described by (5) can be still regarded as a one-dimensional monoatomic mass-spring chain, similar to the one discussed in ref. [24]. Hence, the equation of motion has the form of the Eq. (1), except that the mass terms now include a contribution accounting for the moment of inertia of the disk-shaped atoms.

Therefore, the introduction of chiral elements, made possible by the shape and finite size of the atoms of the phononic crystal, creates additional design elements, such as the angle $\psi$, which

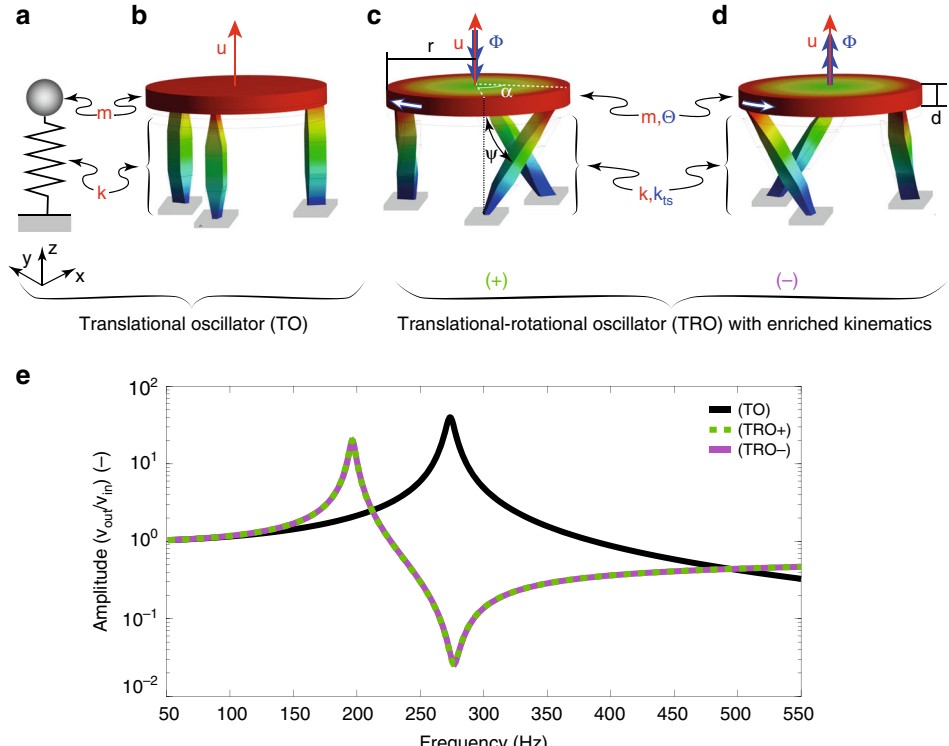

**Fig. 2** Dynamic behavior of ordinary and enriched kinematics oscillators. **a, b** Schematics and three-dimensional model of TO with mass $m$ and axial stiffness $k$. The oscillator can translate in the $z$-direction but cannot rotate, in agreement with the idea that atoms are point masses. **c, d** TROs with enriched kinematics transcending the Haüy-Boscovich model, thanks to their finite extension. Non-centrosymmetric links couple the linear motion ($u$) of the disks along the translational direction ($z$) to their rotation ($\Phi$) about the same axis. The colors in Fig. 2b-d refers to the magnitude of the displacement at the first resonance of the structures, with blue being the smallest and red the largest total displacement, respectively. **e** Amplitudes of the vertical displacement as a function of frequency resulting from a numerical forced frequency response analysis for the three structures reported in Fig. 2b-d. Refer to Supplementary Note 2 for a TRO frequency response up to higher frequencies. Animations of the modes are available online in Supplementary Movies 1–3

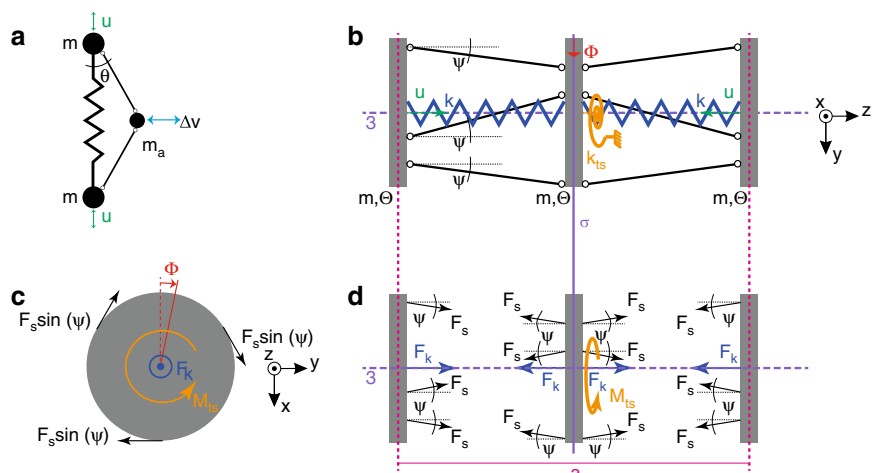

**Fig. 3** Coupling mechanism. **a** Inertial amplification mechanism initially proposed in[23], where the range of motion $\Delta v$ of $m_a$ is a multiple of the range of motion $u$ of $m$: $\Delta v = u/\tan\theta$, due to the kinematic constraint imposed by the stiff levers. **b** Example of a phononic crystal with coupled longitudinal rotational degrees of freedom, in which to a displacement $u$ of the outer atoms corresponds a rotation $\Phi$ of the middle atom. This coupling is enforced by the excentric diagonal struts. The elastic elements, a longitudinal and a torsional spring, with the stiffness $k$ and $k_{ts}$, respectively, provide the restoring force and moment respectively to return the atoms to the equilibrium position. **c, d** Forces and moments acting on the atoms at equilibrium

controls the ratio between the longitudinal and rotational motion ($u$ and $\Phi$), and the ratio of moment of inertia $\Theta$ and mass $m$, that can be exploited to tailor the propagation of mechanical waves. Similarly, also the symmetry operations with respect to the atoms of the unit cell represent an additional tailoring property.

**Concatenation of chiral centers and tacticity**. So far, structured materials with mono-dimensional periodicity have relied on the concatenation of mass-spring oscillators[24], occasionally including chiral elements[40,41]; however, lacking spin-spin coupling, contrary to later works[28,30] where this effect was present but not

specifically studied. Up to now, the assembly strategy has been driven by a simple translation of the unit cell, whereas the concatenation of non-centrosymmetric elements into $1D$ objects has been widely studied in the framework of polymer chemistry with respect to substituted poly-olephines, such as polypropylene. The critical effect of the assembly on the physical properties on otherwise chemically identical polymers has been proved[42,43]. In this context, the notion of tacticity was introduced as the relative stereochemistry of chiral elements. Inspired by this concept, we show the influence of tacticity on the dynamic behavior of chiral structured periodic media, by contrasting the dispersion properties of iso- and a syndiotactic crystals, originating from the different natures (Cosserat and Cauchy, respectively) of the two arrangements.

To this end, we consider two phononic crystals, made of periodic diadic arrays of the TROs reported in Fig. 2c, d in isotactic (Fig. 4a) and syndiotactic (Fig. 4b) arrangement, that correspond to two point groups, namely $\frac{3}{m}$ and $3_1$ (in Hermann–Mauguin notation). In analogy to a polymer (top panels of Fig. 4a, b), the constitution of the building blocks of the two structures is identical, in both cases periodically arranged in the $z$-direction with the same lattice parameter $a$ and only differ for the configuration of the chiral elements (stereoisomers), $(+)/(+)$ and $(+)/(-)$, respectively. Their band diagrams are reported in Fig. 4c, d and are calculated by imposing Bloch-Floquet periodic boundary conditions over the top and bottom atom surfaces and varying the reduced wave number $\mathbf{k}^* = \mathbf{k}_z \cdot \frac{\pi}{a}$ along the $\Gamma - X$ boundary of the first irreducible Brillouin zone (see Methods for further details).

To better understand the nature of the calculated modes, the dispersion curves are color-coded based on a polarization coefficient $p$[44] that quantifies the absolute value of the average $z$-component of the curl of the displacement field (i.e., the rotation about the $z$-axis) of the atoms, representative of the enriched rigid body kinematics of the structures reported in Fig. 2c, d. The color bar representing the polarization factor varies from 0 (blue), indicating that the deformation is localized within the struts, mainly subjected to flexural deformation or as local disk modes, to 400 (red), characterized by a predominantly rigid body motion (rotation + translation) of the disks. This allowed us to identify the modes preeminently involving local deformations (resulting from the continuous nature of the investigated system, blue curves) and those activating the rigid atom rotation in the deformation mechanism of the unit cell (typical of the chiral behavior, the remaining curves). Among the latter, we observe that in the isotactic arrangement, the rigid body rotation of its atoms decreases as the reduced wave number $\mathbf{k}^*$ increases, whereas the syndiotactic structure increases the polarization factor $p$ (the color shading of the bands goes from dark blue to light green) as $\mathbf{k}^*$ increases. Inspecting the mode shapes of the two structures, we observe that the tacticity strongly influences the phase of the rotation of the atoms. In the first case (isotactic arrangement), we can recognize the typical behavior of mass-spring chains[24], where the atoms move in phase along the acoustic branch (A1 in Fig. 4c) and switch to out of phase after their folding at the $X$ point of the edge of the Brillouin zone (optical mode O1 in Fig. 4c). However, this is no longer true, in the case of syndiotactic arrangement, where the top and bottom

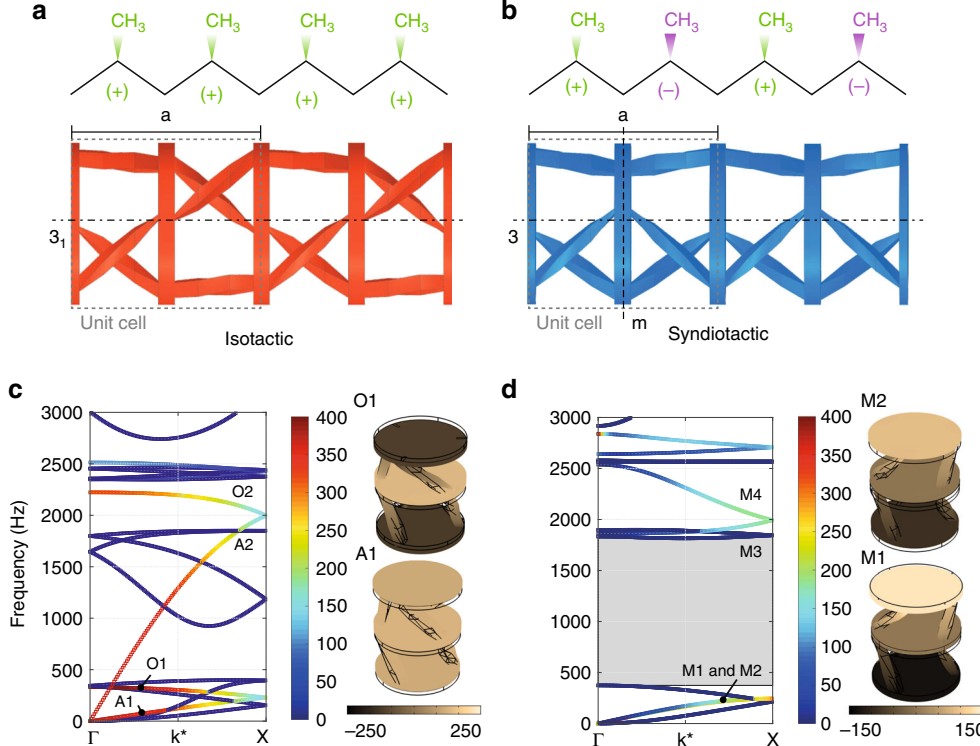

**Fig. 4 Influence of the tacticity. a, b** Schematic representation of the concept of tacticity introduced as the relative stereochemistry of chiral elements in the case of the polypropylene (top panels) inspiring the design of two periodic diadic arrays of TROs in isotactic and syndiotactic configuration (bottom panels). The two structures differ for the tacticity of the non-centrosymmetric elements. **c, d** Band diagrams for the isotactic and syndiotactic configuration. The dispersion curves are color-coded based on a polarization coefficient that quantifies the rigid motion (rotation + translation) about the $z$-axis of the finite size atoms. A large bandgap between 374 Hz and 1816 Hz in the syndiotactic arrangement is visible and highlighted as a light gray rectangle. Higher frequency modes A2, O2, M3, M4 are reported in Supplementary Note 5 and shown in Supplementary Fig. 9

disks are always in opposition of phase leaving the central one still (M1 and M2 in Fig. 4d). This is a direct consequence of the reversed twist of the ligaments connecting the central disk, implying opposite relative displacement of neighboring disks and torsional stress associated with it. This allows the opening of a large bandgap between 374 Hz and 1816 Hz in the syndiotactic arrangement implying the absence of any propagative mode in this frequency range, contrary to the isotactic case. The key idea is that the syndiotactic arrangement of the atoms leads to alternately stationary (with respect to rotation) and moving disks, whereas in the isotactic one, rotation is accumulated along the $z$-axis, breaking the Cauchy media requirements. In other words, in the case of isotactic crystals, under static loads and for a given strain level, the maximum absolute twist will increase proportionally to the distance from the fixed boundary condition, as a function of the number of unit cells, as quantitatively shown in Fig. 5a, c. For the sake of clarity, it is useful to mention that systems such those discussed in the seminal work of Frenzel et al.[31] and the isotactic phononic crystal treated in the present manuscript, cannot be considered a Cauchy continuum. As a consequence, boundary conditions also influence the static properties of these systems. Specifically, if the rotations at their edges are blocked, the stiffness will be higher than if the rotational degrees of freedom are not. On the other hand, the situation is different for the syndiotactic crystal, as the rotation can be considered 'concentrated' in every other disk also under static loads (see Fig. 5b, c). As such, the static stiffness of the material does not depend on the boundary conditions.

It should be pointed out that in this work, the presented band diagrams are calculated through the Bloch-Floquet formalism, which considers the material of infinite extension (i.e., no boundary conditions are given).

## Discussion

In conclusion, we showed that parting from the notion of atoms as point mass elements and embracing the fact that, in phononic crystals, atoms are bound to have a finite size and can thus be attributed a shape, have demonstrably a dramatic effect on the wave propagation properties. This step is a fundamental leap from the idealized model of natural crystals. The introduction of inertia terms, originating from the shape of atoms, in the dispersion equation alone allows to break the otherwise indissoluble link between wave velocity and atomic mass, with obvious practical implications for the phononic materials community. The finite geometry extension of the atoms leads to the definition of (i) coupling ratios between kinematic degrees of freedom (linear + rotational), and (ii) an inertial multiplication factor that determines the effect of inertia on wave velocity. Finally, the introduction of the concept of tacticity in the concatenation of chiral elements adds an additional layer of architecture allowing for the conception of material variants with substantially differing dynamical properties but with the same density and quasistatic stiffness, reminiscent of the differentiation between isotactic and syndiotactic polymers[42,43]. Specifically, beyond the exploitation of inertial effects, the relative orientation of adjacent chiral centers allows to strongly affects the nature of the coupling between the spins of the atoms. The consequent radically different dynamic behaviors, including the nucleation of low frequency full band-gaps, makes tacticity a powerful design strategy of interest in all the fields where vibrations play a crucial role, such as for instance civil, aerospace and mechanical engineering.

## Methods

**Calculation of the frequency response of finite systems**. Mode shapes showing the different behaviors of the achiral and chiral oscillators and the diagram of vertical displacement presented in Fig. 2b–e are calculated by means of ANSYS

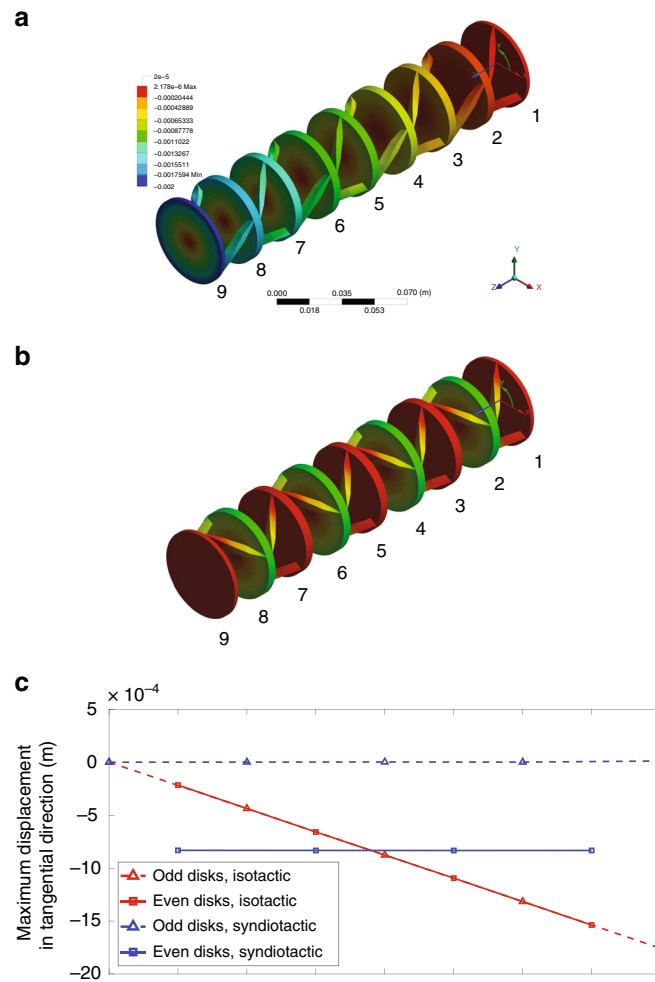

**Fig. 5** Influence of the number of unit cells on the twist for the isotactic and syndiotactic crystals. Surface plots of the $xy$ component of the deformation for the (**a**) isotactic and (**b**) syndiotactic crystals (4 unit cells) subjected to an axial static load. **c** The twist magnitude as a function of the number of disks for the isotactic (red lines) and syndiotactic (blue lines) crystals. The twist increases proportionally to the number of unit cells composing the structure in the isotactic crystal differently from the syndiotactic case

v19.2 as a result of a numerical forced frequency response analysis. The excitation configuration is the same for the three systems and the force is applied at the center of the disks.

The numerical transmission spectra of the isotactic and syndiotactic crystals shown in Fig. 1b, c are calculated using ANSYS v19.2. The elastic properties of the constituting material (DuraForm HST SLS) were estimated from a best match of the numerical and experimental results for the isotactic and syndiotactic crystal, respectively. This approach was used to account for effects on the material properties due to the combination of geometry and manufacturing process. The material properties for the calculations used in Fig. 1b, c were the same as for the calculation of the dispersion curves (Fig. 4c, d) and the static stiffness (Supplementary Fig. 2).

The properties of the struts of the TO of Fig. 2b were divided by a factor of ≈100 to match the static stiffness of the TROs reported in Fig. 2c, d. These modifications are justified by the general nature of the considerations presented in Fig. 2.

**Calculation of the dispersive properties of materials**. Dispersion diagrams and mode shapes presented in Fig. 4c, d are computed using the Bloch-Floquet theory in full 3D FEM simulations, carried out via the Finite Element solver COMSOL Multiphysics. Full 3D models are implemented to capture all possible wave modes. Material densities $\rho_{M,syndio} = 1200 \, \text{kg} \cdot \text{m}^{-3}$ and $\rho_{M,iso} = 1251 \, \text{kg} \cdot \text{m}^{-3}$, based on the measured mass and volume of the phononic crystals shown in Fig. 1a, are assumed. The following orthotropic stiffness matrices and Poisson's ratios were

assumed for the numerical investigation, respectively:

$$\underline{\underline{\mathbb{C}}}_{\text{syndio}} = \begin{bmatrix} 3.03 & 1.14 & 0.86 \\ 1.14 & 3.03 & 0.86 \\ 0.86 & 0.86 & 1.6 \end{bmatrix} \cdot 10^9 \, \text{Pa}, \quad (6)$$

$$\underline{\nu}_{\text{syndio}} = \begin{bmatrix} 0.33 & 0.33 & 0.33 \end{bmatrix}, \quad (7)$$

$$\underline{\underline{\mathbb{C}}}_{\text{iso}} = \begin{bmatrix} 2.74 & 1.06 & 0.82 \\ 1.06 & 2.74 & 0.82 \\ 0.82 & 0.82 & 1.47 \end{bmatrix} \cdot 10^9 \, \text{Pa}, \quad (8)$$

$$\underline{\nu}_{\text{iso}} = \begin{bmatrix} 0.29 & 0.29 & 0.29 \end{bmatrix}, \quad (9)$$

Domains are meshed by means of 10-node tetrahedral quadratic elements of maximum and minimum size $L_{\text{FE}}^{\max} = 3.3$ mm and $L_{\text{FE}}^{\min} = 0.2$ mm, respectively, which allowed for accurate eigensolutions up to the frequency of interest. Mesh refinement was implemented in proximity of the hinge connections. The band structures shown in Fig. 4c, d are obtained assuming periodic conditions along the z-direction. The resulting eigenvalue problem $(\mathbf{K} - \omega^2 \mathbf{M})\mathbf{u} = \mathbf{0}$ is solved by varying the non-dimensional wavevector $\mathbf{k}^* = \mathbf{k}_z \cdot a$ along the boundary of the irreducible Brillouin zone [Γ, X], with $\Gamma \equiv (0, 0)$, $X \equiv (0, \pi/a)$, where $a = 59$ mm is the lattice parameter.

In all the calculations, a linear elastic material assumption is made and geometric non-linearity excluded. Indeed, if geometric non-linearity were at some point triggered, the Bloch-Floquet analysis of the unit cell may loose validity. However, if we consider the typical wave propagation regime for acoustic waves (considering a realistic finite power source of 1 Pa at 500–3000 Hz), the likelihood of triggering geometric non-linearity is minor.

The complex frequency-wave number relations shown in Supplementary Note 4 delivers a straightforward way of investigating the wave attenuation efficiency of metamaterials. The imaginary part of the wave number, calculated by solving a polynomial eigenvalue problem[45,46], quantifies the amplitude decay per meter traveled by the considered wave. The full model was first reduced to a superelement with only 1740 degrees of freedom, chosen for a set of master nodes along the edges of the disks. The stiffness and mass matrices of the superelement have been calculated using the component mode synthesis reduction method[47], readily available in a built-in ANSYS v19.2 routine. We generate the superelement including a truncated set of 125 traction-free eigenmodes, in a frequency range between 50 Hz and 20 kHz.

The resulting dispersion relation predicts the same bandgap as the full model described above, and deviates slightly at higher frequencies due to the model reduction.

**Experimental measurements**. The isotactic and syndiotactic phononic crystals shown in Fig. 1a are manufactured through selective laser sintering (IRPD AG, St. Gallen, Switzerland). The specimens were made of Duraform HST, a commercially available mineral fiber reinforced polyamide, with the following nominal properties: density $\rho_M = 1200$ kg · m$^{-3}$, Young modulus $E_{x,y} = 5.5$ GPa, $E_z = 3$ GPa, and Poisson ratio $\nu = 0.33$. The geometrical parameters are the following: $a = 59$ mm, $r = 25$ mm, $\psi = \approx \pi/4$, $d = 5$ mm.

The experimental data shown in Fig. 1b, c were obtained by using a Polytec PSV 400-H SLDV that measured the out-of-plane velocity of points on a predefined grid (Supplementary Fig. 1) over the structure. The reference input velocity was measured with a Polytec single point LDV (PDV 100) at the edge of the top plate. Elastic waves were excited through a B&K 4801 System V by Brüel & Kjær electrodynamic shaker, driven by the PSV 400. The shaker was screwed to the top surface of the plate at the yellow dot shown in Supplementary Fig. 1. A linear frequency sweep (with harmonic content ranging from 50 Hz to 3000 Hz) lasting 30 s was used as the excitation signal.

The equivalent static stiffness of the two isotactic and syndiotactic phononic crystals under torsional free boundary conditions were experimentally verified as well by performing compression tests in a Zwick Z005 universal testing machine. The experimental setup and the results are shown in Supplementary Fig. 2.

## Data availability

The data (measurement results, code, and models) that support the plots within this paper and other findings of this study are available from the corresponding author upon reasonable request.

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

## Acknowledgements

The authors wish to thank Mr. Marcel Rees and Mr. Hans Michel of the Mechanical Systems Engineering Laboratory, for their invaluable support in performing the static tests on the phononic crystals. Also, the authors wish to thank Mr. Kevin Gebhardt for his invaluable support in the execution of the SLDV transmission measurements of the samples. M.M. is supported by the EMPAPOSTDOCS-II programme, which has received funding from the European Union's Horizon 2020 research and innovation programme under the Marie Skłodowska-Curie grant agreement No. 754364. Finally, the authors wish to acknowledge the funding provided by the Empa Board of Directors under the Internal Research Call scheme (IRC2013) that has made this work possible. The project that lead to this work was funded by Empa under the Internal Research Call scheme. The authors also wish to thank Mr. Sébastien Bergamini for useful discussions.

## Author contributions

A.B. conceived the research, A.B. and M.M. equally contributed to the writing of the manuscript, the formulation of the concept of tacticity in phononic crystals and modal analysis of the iso- and syndiotactic unit cells, T.D. introduced the concept of atoms with rotational moment of inertia and contributed initial experiments, D.T. provided the analytical model of the TRO response. B.V.D. and D.T. contributed the reduced order models for the calculation of the complex wave numbers for the dispersion curves of the crystals., G.H. provided finite element models of the phononic crystals, I.L. contributed the analytical model of the coupling mechanism, A.Z. contributed to the design of experiments and to the writing of the manuscript.

## Competing interests

Certain concepts reported in the present manuscript partially overlap with a patent filed by and granted to Empa (EP 3 239 973). T.D. and A.B. are listed as inventors. The remaining authors declare no competing interests.
