## [Peer Review File · Nature Communications]

Reviewers' Comments:

Reviewer #1:

Remarks to the Author:

The paper introduces and demonstrates a material model that is an improvement to the notion of crystals, typically conceived as ensembles of point masses interacting through attractive and repulsive forces, thereby allowing only translational motions. The classical kinematics of crystals, where coaxial inter-atomic links can oscillate only in one direction, is augmented through the adoption of noncentrosymmetric links that can couple translational to rotational motion, hence sealing into the unit cell chirality with two degree of freedoms. The 3D layout of the unit with a spin and a shift brings about an unusual wave propagation with the formation of band gaps at relatively low frequency. To support the claims, the authors resort to concepts of stereochemistry and classical vibrational analysis with numerical analysis and experiments on 3d printed prototypes.

While there seems to be a merit in this work, a number of points need to be addressed.

1. Material versus structural unit. The unit cell model is proposed to act as primitive for the generation of a one-dimensional periodic lattice material. The demonstration is pursued through the study of a stack of 4 units. Can this be considered a material as claimed in the paper? What is the role of boundary effects enforced on this 4-unit assembly during experiments and simulations? Would the lost of periodicity appear in the response of a limited number of cells? If geometric non-linearity is at some point triggered, the analysis of 1 unit cell can loose validity. These points need further investigation and discussion.

2. Magnitude of twist. What is the relation between the magnitude of twist and the number of stacked units? If less or more units are modelled, I assume the twist would change. A key point to study seems the relation between unit cell number and degree of rotation.

3. Chirality seems one factor controlling the unit cell twist. Chirality has also been long known in the literature, e.g. phononic/photonic and mechanics of periodic materials, and some recent and less recent examples are from Wegener's and Lakes' groups. How does this concept compare with theirs? Although those in the literature are in another physical field, they are yet relevant here? A discussion of this aspect is suggested.

4. 1D bi-atomic mass spring chain. The analysis in Figure S3 and described in the main text led to the normalized bandgap frequency to be between $1/\pi$ and $2/\pi$. The authors claim this to be "raising a fundamental theoretical limitation to the conception of structures with simultaneously high stiffness and low density, but also small unit cell size and low frequency bandgap". But how would their concept compare? I mean how is this limitation made better with their unit cell? I recommend making this evident, and including a figure similar to that given in Fig S3, but this time reporting the performance of their concept.

5. In Figure 3c, the labelled bandgap is shown for an amplitude drop below $\sim 10^{-1}$ (starting from a frequency of $\sim 350\text{Hz}$). However, the amplitude does not rise as much beyond the labelled bandgap with the increase in frequency. Hence the question, why is the bandgap limited to only $\sim 1700\text{Hz}$?

6. Extending from the previous point, in Figure 3b, the amplitude falls to below $\sim 10^{-1}$ as the frequency increases beyond $\sim 2200\text{Hz}$. Does it mean that the isotactic samples should also have a bandgap in that range? There is no explanation of this observation/anomaly. I recommend addressing/discussing this point.

7. Clarification on the derivation of equation 2. Some terms describing the mass and rotary inertia of the struts seem neglected. The authors must justify this in the text.

8. I find the explanation of the coupling mechanism of motions key to the paper. Now it is demoted in the supplementary info (Figure S5). If this explanation is corroborated, I would consider bringing it in the main text, given the main body of the paper is short and three figures only are included. There is opportunity for expansion and corroboration of the results in the main text.

On the presentation, I have also some comments briefly listed here.

- Title. I find the selection of the terms doesn't capture the main gist of the paper. I mean it is not sufficiently descriptive, and from the terms the reader can hardly understand what the context of this "shapely atoms" is. I would recommend alternatives.
- some terms in the text are undefined, e.g. fBG
- Text flow sequence. For nature communications, I would suggest to consider some key descriptive headings to better structure the contents sequence. Currently, everything is shuffled together, making the flow not as streamlined as it could be.
- 'In conclusion, we showed that parting from the notion of atoms as point mass elements and embracing the fact that, in PnCs, atoms are bound to have a finite size and can thus be attributed a shape, has demonstrably a dramatic effect on the wave propagation properties of PnCs.' Change 'has' to 'have'.
- In the supplementary materials, under 'Coupling mechanism description', after equations 6, it should be 'similar to the one'. Meaning add "to"
- c. In the same section as the previous point, the last line talks about ' $3m$ ' but, there seems to be nothing marked as such in Figure 2a,b.
- In Figure S6, should it be 'The acoustic (A2) and optical (O2) modes' in the first line?

Reviewer #2:

Remarks to the Author:

In this manuscript, the authors have proposed a methodology in which by realizing a coupling between torsional and translational oscillation, dynamic properties have been altered. Importantly, it is demonstrated that by leveraging the coupling mechanism, bandgap may be formed below the Bragg scattering limit regardless of the change in stiffness and mass term. Eventually, inspired by the concept of Tacticity in polymer science, two different assembly strategies have been investigated. It is demonstrated that these two assembly strategies with the identical building blocks, yields to the radically different dynamic properties which have further validated the significance of the assembly strategy, i.e., Tacticity. Throughout the manuscript, presented claims and conclusions have been justified with convincing numerical and experimental results.

I hesitate on recommending this submission (its present form) for publication based on one major issue.

A significant portion of the present manuscript (both claims and context) ranging from the abstract to the conclusion and supplementary information is dedicated to the concept of coupled torsional/translational mechanism and its application for low-frequency bandgap. This concept and particularly the behavior of the presented structures have been properly covered in the previously published work of the authors (Inertia Amplification in phononic crystals for low-frequency bandgaps. VIII ECCOMAS SMART 2017. 1657-1668) and other researchers (Ref 36 and Ref 34 of the manuscript: Orta, A. H. & Yilmaz, C. Inertial amplification induced phononic band gaps

generated by a compliant axial to rotary motion conversion mechanism. Journal of Sound and Vibration 439, 329 – 343 (2019)).

I would like to see the manuscript revised based on the above issue with minimal overlap of claims and context with already published works.

Minor Issues and Suggestion:

1- I believe FEM and measured results in Fig. 3b (Isotactic) are not in a good agreement between 400-1500 Hz, this can be partially due to the fact that stiffness tensor has been approximated by fitting the numerical and experimental results in the case of Syndiotactic configuration. I would like to see how the band gap of Syndiotactic configuration will change by instead approximating the stiffness tensor using Fig. 3b (Isotactic).

2- Would be great to include the analytically derived response in addition to the numerically derived one in fig.1 e

3- Refernce#30 needs to be modified.

Response to Reviewers

August 19, 2019

We thank the reviewers for their comments and positive feedback. With this letter we would like to respond to the reviews and summarize changes made. All changes are also highlighted in red in the present document and in the revised manuscript.

In the list of authors, "D. Tallarico" and "B. Van Damme" have been added for their contribution to the revised version of the manuscript. Their specific contribution has been itemized in the "Author contribution section" as follows: D.T. provided the analytical model of the TRO response. B.V.D. and D.T. contributed the reduced order models for the calculation of the complex wavenumbers for the dispersion curves of the crystals.

Reviewer 1

1. **Comment:** *Material versus structural unit. The unit cell model is proposed to act as primitive for the generation of a one-dimensional periodic lattice material. The demonstration is pursued through the study of a stack of 4 units. Can this be considered a material as claimed in the paper? What is the role of boundary effects enforced on this 4-unit assembly during experiments and simulations? Would the loss of periodicity appear in the response of a limited number of cells? If geometric non-linearity is at some point triggered, the analysis of 1 unit cell can lose validity. These points need further investigation and discussion.*

- **Response:** We agree with the reviewer that boundary conditions may alter the overall response of the system in experiments and simulations and that theoretically, only a sample of infinite extension would allow to record the actual *material* properties, as they are represented by the dispersion curves. However, it has been shown that the footprint of the dispersion properties of a unit cell is already appreciable in the behavior of a finite structure, although obfuscated by boundary effects [1]. In reason of this, and supported by both the numerical and experimental measurements of the transmissibility dip (see Fig. 5c of the main text), we believe that the choice of a finite structure spanning over 2 unit cells is sufficient to give a reasonable evidence of the *material* properties. Nevertheless, we carried out supplementary calculations for further finite systems made of 1, 3 and 4 unit cells, respectively, to further strengthen this conclusion. Figure S4 reports the transmission diagram as a function of the number of unit cells (1 to 4). Results show that in the syndiotactic crystal the destructive interference mechanism starts taking place already when only 2 unit cells are present and it can be observed that as the number of unit cells increases, the attenuation dip strengthens and the edges

of the attenuation frequency regions become sharper (see Fig. S4b). The results are presented in semi-logarithmic scale and prove that each additional unit cell introduces an attenuation factor of about 20. On the other hand, in the case of isotactic crystal, where no bandgap is expected in the considered frequency range, we only observe a shift of the structural resonance peaks, due to the increased length of the sample and, accordingly, of the frequencies at which standing waves are observed (see Fig. S4a).

We also agree that if geometric non-linearity is at some point triggered, the Bloch-Floquet analysis of the unit cell may lose validity. However, if we consider a typical wave propagation regime for acoustic waves (considering a realistic finite power source of 1 Pa at 500-3000 Hz), typically, the likelihood of triggering geometric non-linearity is minor. Therefore, the system is modeled as linear elastic, so that the signal attenuation is exclusively due to the Bragg scattering mechanism. As a consequence, we believe that this analysis would go beyond the scope of our work and we did not consider this aspect. However, a remark in this sense has been added in the text.

To clarify the aforementioned aspects, we amended the paper as follows:

- Figure S4 has been added to the supplementary materials. It includes the *calculated* transmissibility spectra for finite structures comprised of 1 to 4 unit cells and shows the trend of the *material* properties of finite structures towards those of an infinite crystal, as well as the imaginary part of the wave number as a function of the frequency.

The following text describing the additional figures have been added into the supplementary materials: **The boundary conditions may alter the overall response of the system in experiments and simulations. Theoretically, only a sample of infinite extension would allow to record the actual material properties, as they are represented by the dispersion curves. However, it has been shown that the footprint of the dispersion properties of a material is already appreciable in the behavior of a finite structure, although obfuscated by boundary effects [1]. In reason of this, and supported by both the numerical and experimental measurements of the transmissibility dip (see Fig. 5c of the main text), we believe that the choice of a finite structure spanning over 2 unit cells is sufficient to give a reasonable evidence of the material properties. To support this statement, we carried out supplementary calculations for further finite systems made of 1, 3 and 4 unit cells, respectively. Figures S4a,b report the transmission diagram as a function of the number of unit cells (1 to 4). Results show that in the syndiotactic crystal (Fig. S4b) the destructive interference mechanism within the bandgap starts taking place already when only 2 unit cells are present and that as the number of unit cells increases the attenuation dip strengthens and the edges of the attenuation frequency regions become sharper. A fit of the transmissibility after 1, 2, 3, and 4 unit cells (i.e., at $x = a = 0.059$ m, $x = 2a$, $x = 3a$ and $x = 4a$, respectively) shows an exponential amplitude decay proportional to $\exp(-34.4x)$ and $\exp(-53.9x)$ for $f = 375$ Hz and $f = 1030$ Hz, respectively.**

On the other hand, in the case of the isotactic crystal, where no bandgap is expected in the considered frequency range, only a shift of the structural resonance peaks is observed, due to the increased length of the sample and, accordingly, of the frequencies at which standing waves are produced (Fig. S4a).

In addition, Figs. S4c,d show the imaginary wave numbers in the isotactic and syndiotactic cases, calculated for all the modes, to estimate the spatial attenuation of the crystals, as a function of the frequency. The imaginary wave number directly relates to the wave attenuation retrieved from the full models. In the case of the syndiotactic crystal (Fig. S4b), solely strongly attenuated waves (i.e. $\Re(k) = n\pi/a$) are present in the range between approximately 370 Hz and 1800 Hz (corresponding to the full bandgap reported in Fig. 3d of the main text). Specifically, in this frequency range, four branches are visible: the lowest one refers to a bending wave not visible in Fig. 3d of the main text (because characterized by $\Re(k) = n\pi/a$). The middle 2 curves belong to longitudinal waves (M1 and M2 of Fig. 3d of the main text), and the highest branch belongs to the M3 mode with a cut-on frequency above 1800 Hz. The imaginary wave number values of these branches perfectly agree with the exponential fit at the same frequencies shown in Fig. S4a,b.

In the isotactic case (Fig. S4c), the A1 mode reaches very high attenuation levels starting from 355 Hz, while the A2 mode exhibits no attenuation up to approximately 2000 Hz, confirming the high transmissibility calculated in the full model and observed experimentally. Only in the frequency range between approximately 2000 Hz and 2400 Hz, we observe a region of modes with purely imaginary wave numbers. However, the imaginary component of the lowest branch (corresponding to A2 mode) assumes values not exceeding 20 m^{-1} .

In conclusion, the results presented in this section confirm the superior efficiency of the syndiotactic crystal in creating low frequency and deep bandgaps for a given mass density and structural stiffness, as compared to the isotactic case, proposed in previous studies [2, 3, 4].

- The following remark about the assumption of geometric and material linearity has been added to the description of the investigated systems: **In all the calculations, a linear elastic material assumption is made and geometric non-linearity excluded.** Indeed, if geometric non-linearity were at some point triggered, the Bloch-Floquet analysis of the unit cell may lose validity. However, if we consider the typical wave propagation regime for acoustic waves (considering a realistic finite power source of 1 Pa at 500-3000 Hz), the likelihood of triggering geometric non-linearity is minor..

2. **Comment:** *Magnitude of twist. What is the relation between the magnitude of twist and the number of stacked units? If less or more units are modeled I assume the twist would change. A key point to study seems the relation between unit cell number and degree of rotation.*

- **Response:** We agree with the Reviewer that, in the case of isotactic crystal, *under static loads* and for a given *strain* level, the maximum absolute twist will increase proportionally to the distance from the fixed boundary condition, expressed in number of unit cells, as quantitatively shown in Figs. 4a,c. For the sake of clarity, it is useful to mention that systems such those discussed in the seminal work [5] and the isotactic phononic crystal treated in the present manuscript, cannot be considered a Cauchy continuum. As a consequence, boundary conditions also influence the static properties of these systems. Specifically, if the rotations at their edges are blocked, the stiffness will be higher than if the rotational degrees of freedom are not. On the other hand, the situation is different for the syndiotactic crystal, *as the rotation can be considered*

'concentrated' in every other disk also under static loads (see Figs. 4b,c). As such, the static stiffness of the material does not depend on the boundary conditions.

It is worth pointing out that in this work the presented band diagrams are calculated through the Bloch-Floquet formalism which considers the material of infinite extension (i.e., no boundary conditions are given).

To address these concerns, we amended the paper as follows:

- In the section '**Experimental measurements**' of the supplementary materials, we highlighted the fact that the mechanical properties of the two crystals were tested under the same (free-free) conditions with respect to the rotation of the end disks.
- Figure 4 has been added to the supplementary materials.
- The following text has been added to the main text: **In the case of isotactic crystal, under static loads and for a given strain level, the maximum absolute twist will increase proportionally to the distance from the fixed boundary condition, expressed in number of unit cells, as quantitatively shown in Figs. 4a,c. For the sake of clarity, it is useful to mention that systems such those discussed in the seminal work of Frenzel et al. [5] and the isotactic phononic crystal treated in the present manuscript, cannot be considered a Cauchy continuum. As a consequence, boundary conditions also influence the static properties of these systems. Specifically, if the rotations at their edges are blocked, the stiffness will be higher than if the rotational degrees of freedom are not. On the other hand, the situation is different for the syndiotactic crystal, as the rotation can be considered 'concentrated' in every other disk also under static loads (see Figs. 4b,c). As such, the static stiffness of the material does not depend on the boundary conditions.**

It should be pointed out that in this work, the presented band diagrams are calculated through the Bloch-Floquet formalism which considers the material of infinite extension (i.e., no boundary conditions are given).

3. **Comment:** *Chirality seems one factor controlling the unit cell twist. Chirality has also been long known in the literature, e.g. phononic / photonic and mechanics of periodic materials, and some recent and less recent examples are from Wegener's and Lakes' groups. How does this concept compare with theirs? Although those in the literature are in another physical field, they are yet relevant here? A discussion of this aspect is suggested.*

- **Response:** We thank the Reviewer for this stimulating comment, as it allows us to add perspective to our work. *Chirality and the relative arrangement of chiral centres along a chain (namely, the tacticity of the crystal) are the central issues to this work. Indeed, the tacticity of a crystal allows to switch from a Cauchy- to a Cosserat-like material, by applying specific symmetry operations, while maintaining the same properties in terms of density and stiffness, under appropriate boundary conditions.*

In this context, we found the following works by Wegener's and Lakes' Group central to our work and critical to clearly trace the analogies and main differences between the added value of chirality within phononic and photonic domains, avoiding concept repetitions and at the same time allowing to emphasize our novelty, which relies in the exploration of the differences in the dispersive properties arising from the *contrast between an isotactic and a syndiotactic unit cell arrangement*, leading to a micropolar and Cauchy behavior of phononic crystals, respectively.

- References [6, 7, 8, 9] (added to the reference list of the new version of the manuscript), which allowed us to relate our results to relevant work in the field of phononics and photonics.
 - Reference [5] (already cited in the previous version of the manuscript) and reference [10] (added), where the effect of the hinge stiffness on the wave propagation is discussed;
 - Reference [11] (added) and reference [12] (added), which focus on the quasi-static properties of micropolar materials and on the dispersive properties of continua, respectively.
- To highlight this, we added the following sentences to the paper:

- The concept of chirality has been long known in the literature and while it has been largely explored in the field of electromagnetic/optical metamaterials, its potential only recently emerged in elasticity [6, 7, 8, 10]. Although substantial differences between the two domains (the absence of static chiral effects in optics and the mass density and mass density tensor additionally entering via the equation of motion in the elastic case) do not allow a direct transposition of concepts explored in one field to the other, a close mathematical analogy at the level of effective-medium description allows cross-fertilization inspiring new physics, such as negative refractive indices [9] or opening the path to the exploration of quasi-static properties of micropolar materials [11] and the dispersion relation of continua [12], among others.
- [...periodic media], by contrasting the dispersion properties of iso- and a syndiotactic crystals, originating from the different natures (Cosserat and Cauchy, respectively) of the two arrangements.

4. **Comment:** *1D bi-atomic mass spring chain. The analysis in Figure S3 and described in the main text led to the normalized bandgap frequency to be between $1/\pi$ and $2/\pi$. The authors claim this to be "raising a fundamental theoretical limitation to the conception of structures with simultaneously high stiffness and low density, but also small unit cell size and low frequency bandgap". But how would their concept compare? I mean how is this limitation made better with their unit cell? I recommend making this evident, and including a figure similar to that given in Fig S3, but this time reporting the performance of their concept.*

- **Response:** The objective of 'breaking' the limit imposed by the ω^* relation is the original motivator of the work. While a parametric optimization to design stiff, lightweight materials with simultaneously low frequency bandgap is always possible, the main objective here is to show the remarkable influence of the symmetry operations used to build the unit cell of a chiral phononic crystal on its dynamic properties. Therefore, in order to allow for a fair comparison of the iso- and syndiotactic version of a phononic crystal with identical base units, we had to impose an additional geometric constraint to the design of the unit cell. In order to guarantee proper translational periodicity, the angle α (see modified Fig. 1 in the revised manuscript) between the projections of the intersection of the links with the disk-shaped atoms had to be $\alpha = \pi/3$. This ensured that upon translation of the unit cell, neighboring links would be positioned adjacent to each another, both in the isotactic and the syndiotactic configuration. This choice allowed for a resulting value of ω^* , as calculated from the numerical models (to exclude any experimental error), to be below the described limit for monoatomic point mass PnCs, namely $2/\pi = 0.637$.

For the sake of completeness, the ω^* for the syndiotactic phononic crystal presented in the main text is calculated according to the inline equation shown on page 3 of the manuscript $\omega^* = f_{BG}a\sqrt{\frac{\rho}{C}}$, with $f_{BG} = 374$ Hz, see Fig. 3d, $a = 0.059$ m, $\rho = 203$ kg/m³, $C = 1$ MPa leading to $\omega^* = 0.314$ (with $2/\pi = 0.637$ for monoatomic crystals).

We highlighted this in the SM of the paper by adding the following sentence:

- For the sake of completeness, the ω^* for the syndiotactic phononic crystal presented in the main text is calculated according to the inline equation shown on page 3 of the manuscript $\omega^* = f_{BG}a\sqrt{\frac{\rho}{C}}$, with $f_{BG} = 374$ Hz, see Fig. 3d, $a = 0.059$ m, $\rho = 203$ kg/m³, $C = 1$ MPa leading to $\omega^* = 0.314$ (with $2/\pi = 0.637$ for monoatomic crystals).
5. **Comment:** *In Figure 3c, the labelled bandgap is shown for an amplitude drop below 10-1 (starting from a frequency of 350Hz). However, the amplitude does not rise as much beyond the labelled bandgap with the increase in frequency. Hence the question, why is the bandgap limited to only 1700Hz?*

- **Response:** The indication of the band gap in Fig. 5c (formerly Fig. 3c) is based on the calculated dispersion curves shown in Fig. 3d. The edges of the band gap are 374 Hz at the bottom and 1833 Hz at the top. We changed the text from 'Band gap' to '**Calculated Bandgap**' and we completed the caption of Fig. 3d with the following sentence: **visible and highlighted as a light gray rectangle**.

We understand the Reviewer's concern about the low amplitudes registered in the high frequency range of the measured and calculated results. In the submitted manuscript we did not mention the assumptions we made for the damping properties of the system. Given the nature of the material/structure (selectively laser sintered polymer), we assumed a damping coefficient for the whole structure, estimated at $c=2.5E-2$, based on a reasonable amplitude match between model and experiment. Under these conditions, the higher frequency resonances are fairly strongly damped, as observed in the measurements (gray line in Fig. 5b,c). However, we performed additional numerical calculations parametrically changing the value of the material damping. Fig. R1 shows that for a very weakly damped system, the amplitude response can indeed be expected to raise above 10^0 .

6. **Comment:** *Extending from the previous point, in Figure 3b, the amplitude falls to below 10-1 as the frequency increases beyond 2200Hz. Does it mean that the isotactic samples should also have a bandgap in that range? There is no explanation of this observation / anomaly. I recommend addressing / discussing this point.*

- **Response:** We agree with the Reviewer's observation that the transmittance of the isotactic crystal falls below 10^{-1} at frequencies above 2200 Hz (i.e. above the O2 mode). This is due to the band structure of the isotactic crystal (see Fig 3c) that presents few flat modes with small gaps in between them. Further, as far as the finite element calculation of the transmission curve, it should be noted that a very simple damping model (proportional to the stiffness matrix) has been used. Consequently an attenuation proportional to frequency is expected.

We introduced this information in the caption of Fig. 5 as follows:

- The transmittance of the isotactic crystal falls below 10^{-1} at frequencies above 2200 Hz (i.e. above the O2 mode). This is due to the band structure of the isotactic crystal (see Fig 3c) that presents few flat modes with small gaps in between them.
- It should be noted that a very simple damping model (proportional to the stiffness matrix) has been used for the finite element calculation of the transmission curves. Consequently, an attenuation proportional to the frequency is expected. The lines in strong color (dark red, dark blue) refer to the frequency response calculated based on the fit from the respective crystal, while the light color indicates the curve calculated using the properties obtained from the other crystal. While the position of features moves slightly, the character of the curves does not change.

7. **Comment:** *Clarification on the derivation of equation 2. Some terms describing the mass and rotary inertia of the struts seem neglected. The authors must justify this in the text.*

- **Response:** The Reviewer’s observation is correct. No mass or inertia is accounted for the links between the disks, as their only function is to impose the *kinematic* coupling (longitudinal-torsional) between them. Just like in the treatment of the mass-spring chain presented in the work of Brillouin, we chose to idealize the system and assume that the spring and coupling elements be massless. However, we agree that more realistic models may represent elasticity and kinematics of the system using beam elements, like the analytical TRO model we developed upon suggestion of Reviewer 2, and reported in the supplementary materials.

8. **Comment:** *I find the explanation of the coupling mechanism of motions key to the paper. Now it is demoted in the supplementary info (Figure S5). If this explanation is corroborated, I would consider bringing it in the main text, given the main body of the paper is short and three figures only are included. There is opportunity for expansion and corroboration of the results in the main text.*

- **Response:** We agree with the reviewer on this point and we moved the section in question to the main text.

9. **Comment:** *On the presentation, I have also some comments briefly listed here.*

- Title. I find the selection of the terms doesn’t capture the main gist of the paper I mean it is not sufficiently descriptive, and from the terms the reader can hardly understand what the context of this ”shapely atoms” is. I would recommend alternatives.

Response: The title has been extended to ”Shapely Atoms: Tacticity in Chiral Phononic Crystals”

- some terms in the text are undefined, e.g. fBG

Response: We have carefully searched through the text and defined the missing terms when they first appear in the text.

- Text flow sequence. For nature communications, I would suggest to consider some key descriptive headings to better structure the contents sequence. Currently, everything is shuffled together, making the flow not as streamlined as it could be.

Response: we have introduced 5 headings to better structure the text.

- 'In conclusion, we showed that parting from the notion of atoms as point mass elements and embracing the fact that, in PnCs, atoms are bound to have a finite size and can thus be attributed a shape, has demonstrably a dramatic effect on the wave propagation properties of PnCs.' Change 'has' to 'have'.

Response: suggested change made

- In the supplementary materials, under 'Coupling mechanism description', after equations 6, it should be 'similar to the one'. Meaning add "to"

Response: suggested change made

- In the same section as the previous point, the last line talks about '3m' but, there seems to be nothing marked as such in Figure 2a,b.

Response: 'm' refers to the mirror symmetry plane indicated with a dashed line in figure 3b. The dashed line divides the first full 'atom' (as counted from the left). The letter 'm' is to the left of the word 'syndiotactic'. The notation ' $\frac{3}{m}$ ' indicates a threefold symmetry axis perpendicular to a mirror plane, using the HermannMauguin notation.

- In Figure S6, should it be 'The acoustic (A2) and optical (O2) modes' in the first line?

Response: Figure amended.

Reviewer 2

1. **Comment:** *I hesitate on recommending this submission (its present form) for publication based on one major issue. A significant portion of the present manuscript (both claims and context) ranging from the abstract to the conclusion and supplementary information is dedicated to the concept of coupled torsional / translational mechanism and its application for low-frequency bandgap. This concept and particularly the behaviour of the presented structures have been properly covered in the previously published work of the authors (Inertia Amplification in phononic crystals for low-frequency bandgaps. VIII ECCOMAS SMART 2017. 1657-1668) and other researchers (Ref 36 and Ref 34 of the manuscript: Orta, A. H. & Yilmaz, C. Inertial amplification induced phononic band gaps generated by a compliant axial to rotary motion conversion mechanism. Journal of Sound and Vibration 439,329 - 343 (2019)). I would like to see the manuscript revised based on the above issue with minimal overlap of claims and context with already published works.*

- **Response:** While we agree with the Reviewer that the exploitation of the rotational inertia of atoms has been discussed to some extent elsewhere (including a conference

proceeding by some of the authors of this manuscript) the core of the matter discussed here (the effect of *tacticity* on the dispersive properties of spin-coupled phononic crystals) has not been discussed anywhere else, to the best of our knowledge. For this purpose, in this work, we have isolated the functional elements needed to demonstrate this concept, including: tacticity and the effect of rotational inertia. In the spirit of offering the Reader a complete picture, both points have been presented.

Concerning the overlap of the topics treated here and in previous work by other authors, we believe that the explanation of the coupling mechanisms of motion is key to putting the work into context, especially given the fact that these concepts are far from being widely known. The perceived general lack of perspective on the relationship between bandgap position and relevant engineering properties of materials (such as stiffness and mass density) needs to be highlighted to the Reader. We believe that only with this perspective in mind, the significance of the exploitation of the inertia of atoms and the effects of tacticity for the stated purposes can be fully appreciated.

2. **Comment:** *I believe. FEM and measured results in Fig. 3b (Isotactic) are not in a good agreement between 400-1500 Hz, this can be partially due to the fact that stiffness tensor has been approximated by fitting the numerical and experimental results in the case of Syndiotactic configuration. I would like to see how the band gap of Syndiotactic configuration will change by instead approximating the stiffness tensor using Fig. 3b (Isotactic).*

- **Response:** We agree with the reviewer that the match between the measured and calculated Response function for the isotactic crystal is not as good as for the syndiotactic one. The presence of a strong band gap (see Fig. S4b) in the syndiotactic crystal made it easy to update the material properties to account for any deviations and inaccuracies (due to geometry or to the effect of the manufacturing process on the properties of the material in different parts of the crystal) occurring in the manufacturing process of the samples. For the isotactic crystals, however, we do not have clear landmarks in the frequency response as we do in the spectra for the syndiotactic crystal, so that updating the material properties based on the frequency response is not as straight forward as for syndiotactic crystal.

So, based on the fact that:

- the two samples were made of the same exact material
- the samples were produced in the same run
- the detail geometries of the two crystal only differ by a mirror symmetry plane in the mid plane of the disks
- the quasistatic properties of the two samples are essentially identical and show a good match between experimental and numerical results,

we did not expect strong differences between the material properties of the two samples. So, we chose to use the same material properties (as updated based on the syndiotactic crystal) for all models, at the cost of having a not so nice match for the dynamic response of the isotactic crystal. This does not in any way affect the core statement of the manuscript that tacticity in chiral phononic materials has a profound effect on their dispersive properties, while it does not necessarily affect other physical properties. However, based on this very argument, nothing stands in the way of updating the

material properties based on the isotactic frequency response function and making the comparison between isotactic and syndiotactic crystals, using slightly different elasticity parameters. In Fig. 5, the calculated frequency response functions of the crystals are reported as follows:

- strong color (red for isotactic, blue for syndiotactic) for the case where the material properties are updated based on the respective experiment,
- pale color for the case where the material properties are updated based on the experimental results from the other crystal

The frequency response functions reported in the supplementary materials for additional discussion are calculated based on the properties as fitted with the respective experiments (strong color).

3. **Comment:** *Would be great to include the analytically derived response in addition to the numerically derived one in fig.1 e*

- **Response:** We agree with the reviewer that in general, analytical models are a useful way to represent mechanical systems, even if in this specific case, numerical model well represents the behavior of the system. However, we have prepared an analytical model of the TRO, discussed in Fig. 1 of the manuscript and reported on it in the supplementary materials. Furthermore, we refer to an available analytical model of the coupling in a component of a similar system, available in literature add here reference to Yilmaz model. The complex geometry of the trusses in the discussed crystal would lead to an extremely cumbersome analytical model, with respect to the representation of their elastic properties. Since this complexity does not add to the understanding of the effect of including linear-torsional coupling of the motion of the atom, we have decided to present a simplified model with long slender trusses, that we model as Euler-Bernoulli beams.

We therefore added a section in the supplementary materials entitled "Analytical model of the TRO", reported below for the sake of clarity:

Analytical model of the TRO

In this section, we formulate and solve an analytical model for a single TRO with the aim of (i) capturing the basic physics underlying its resonance and antiresonance phenomena and (ii) predicting the (low-)frequency response function (frf) in a closed analytical form.

As presented in the main text, the conceived model of the TRO includes deformable elastic elements such as tapered and twisted beams, tilted with respect to the vertical direction of an angle Ψ , connected to disks (also deformable).

A detailed analytical description of such a multi-structure can be derived using asymptotic techniques [13, 14]. Here, to obtain the (low-)frequency response function (frf) of a TRO in a closed analytical form, we assume that (i) the disks are rigid (i.e., of infinite stiffness), (ii) the elastic properties of the beams isotropic and (iii) their cross-sections of constant circular shape throughout their longitudinal coordinate. This allowed the modeling of the beams

according to the Euler-Bernoulli theory. The tilting of the beams with respect to the vertical coordinate has been retained so to point out its effect on the frf of a single TRO, as the one schematically represented in Fig. S7, where the three Euler-Bernoulli beams are represented by the dashed lines. Each of them, *e.g.* the one marked in red, supports a displacement vector of the form:

$$\mathbf{r}(x_1, t) = w(x_1, t)\mathbf{e}_2 + v(x_1, t)\mathbf{e}_1. \quad (1)$$

The transverse w and longitudinal v displacements of the beam are governed by two decoupled PDEs [15]:

$$\begin{aligned} c^2 \frac{\partial^2}{\partial x_1^2} (v(x_1, t)) - \frac{\partial^2}{\partial t^2} (v(x_1, t)) &= 0, \quad x_1 \in [0, \ell], \\ c^2 \mathcal{R}^2 \frac{\partial^4}{\partial x_1^4} (w(x_1, t)) + \frac{\partial^2}{\partial t^2} (w(x_1, t)) &= 0, \quad x_1 \in [0, \ell], \end{aligned} \quad (2)$$

for the longitudinal and flexural fields, respectively. In Eqs. (2) we have introduced the longitudinal wave speed $c = \sqrt{E/\rho}$, with E being a homogeneous Young's modulus and ρ the mass density, $\mathcal{R} = \sqrt{\mathcal{I}/S}$ the gyration radius of the beams, S their cross section and \mathcal{I} their second moment of inertia.

Kinematic constraints on the beams

It is useful to introduce here three three-dimensional coordinate systems (CSs), as shown in Fig. S7: $(\mathbf{e}_x, \mathbf{e}_y, \mathbf{e}_z) \equiv \text{diag}(1, 1, 1)$ and $(\mathbf{e}_1, \mathbf{e}_y, \mathbf{e}_2)$, with $\mathbf{e}_1 = R(\Psi, \mathbf{e}_y)\mathbf{e}_x$ and $\mathbf{e}_2 = R(\Psi, \mathbf{e}_y)\mathbf{e}_z$, having denoted $R(\theta, \mathbf{e})$ as the anti-clockwise rotation matrix of angle θ around a unit vector \mathbf{e} . The third CS $(\mathbf{e}_3, \mathbf{e}_\theta, \mathbf{e}_z)$ is such that $\mathbf{e}_\theta = R(\Delta/2, \mathbf{e}_z)\mathbf{e}_x$ and $\mathbf{e}_3 = R(\Delta/2, \mathbf{e}_z)\mathbf{e}_y$, being:

$$\Delta = 2 \arcsin(\cos(\Psi) \frac{r}{2\ell}), \quad \text{with} \quad r \leq \frac{2\ell}{\cos(\Psi)}. \quad (3)$$

We assume that the TRO in Fig. S7 vibrates as a result of a time-harmonic prescribed displacement applied to the base of the beams $x_1 = 0$,

$$\mathbf{u}_0(t) = u_0 e^{-i\omega t} \mathbf{e}_z.$$

The displacement field in Eqs. (1) is therefore time-harmonic of frequency ω , *i.e.*:

$$w(x_1, t) = \bar{w}(x_1) e^{-i\omega t}, \quad v(x_1, t) = \bar{v}(x_1) e^{-i\omega t}. \quad (4)$$

Moreover, we introduce the scaled variables:

$$\bar{x}_1 = x_1/\ell, \quad W(\bar{x}_1) = \bar{w}(\ell\bar{x}_1)/u_0, \quad V(\bar{x}_1) = \bar{v}(\ell\bar{x}_1)/u_0, \quad (5)$$

where u_0 is the amplitude of the time harmonic external excitation. Using Eqs. (4) and (5), and focusing on the low-frequency regime, we can rewrite the PDEs (2) as ODEs:

$$V''(\bar{x}_1) = 0, \quad \text{and} \quad W''''(\bar{x}_1) = 0, \quad \bar{x}_1 \in [0, 1], \quad (6)$$

where we have introduced $(\cdot)' \equiv \partial/\partial x_1(\cdot)$, whose general solutions are:

$$V(\bar{x}_1) = V_0 + V_1\bar{x}_1, \quad \text{and} \quad W(\bar{x}_1) = W_0 + W_1\bar{x}_1 + W_2\bar{x}_1^2 + W_3\bar{x}_1^3. \quad (7)$$

The motion of a rigid disk in Fourier transform comprises a vertical displacement and a rotation around the \mathbf{e}_z axis, *i.e.*:

$$\mathbf{U}(\omega)/u_0 = U_{\text{out}}(\omega)\mathbf{e}_z + \Theta_{\text{out}}(\omega)\mathbf{e}_\theta. \quad (8)$$

In Eq. (8) we have introduced the normalized vertical displacement amplitude of the disk as $U_{\text{out}} = \bar{U}_{\text{out}}/u_0$ and its normalized angular displacement around \mathbf{e}_z , $\Theta_{\text{out}} = \bar{\Theta}_{\text{out}}r/u_0$, being r the radius of the disk.

In a regime of small deformations, the following set of boundary conditions apply to the longitudinal displacement field of the beam (first relation in Eqs. (7)):

$$V(0) = \mathbf{e}_z \cdot \mathbf{e}_1, \quad V(1) = (\Theta_{\text{out}}\mathbf{e}_\theta + U_{\text{out}}\mathbf{e}_z) \cdot \mathbf{e}_1, \quad (9)$$

whereas the ends conditions for the flexural displacement field (second function in Eqs. (7)) are:

$$\begin{aligned} W(0) &= \mathbf{e}_z \cdot \mathbf{e}_2, \quad W(1) = (\Theta_{\text{out}}\mathbf{e}_\theta + U_{\text{out}}\mathbf{e}_z) \cdot \mathbf{e}_2, \\ W'(0) &= 0, \quad W'(1) = 0. \end{aligned} \quad (10)$$

The simplified set of boundary conditions for the flexural motion (10) is compatible with those used by Orta and Yilmaz [4]. However, in our model we also take into account the force \mathbf{F} (see Fig. S7), which results from the longitudinal deformation of the beams, which was on the contrary neglected by Orta and Yilmaz [4]. It is worth mentioning here that although the longitudinal force \mathbf{F} plays no role on the dynamics of the disk when the beams are tilted, it significantly contributes to the dynamics of the disk at $\Psi = \pi/2$ (*i.e.*, for vertical beams). Therefore in our analytical model we have retained the longitudinal forces due to the elongation of the beams to avoid limitations on Ψ values the model can treat. A related problem was described by Tallarico *et al.* [16] in the contest of a two dimensional mass-truss lattice with its unit cell containing a tilted resonators. In this work, it is shown that the structure is degenerate - *i.e.* possesses a vanishing torsional frequency - at zero tilting angle and that such degeneracy can be cured introducing flexural ligaments. For these reasons, we believe that accounting for *both* flexural and longitudinal reaction forces to the supporting beams is of pivotal importance to obtain consistent analytical models of geometrically chiral and tactic structures.

Frequency response functions of a single TRO

The equations for the time-harmonic motion are readily obtained by the balance of linear momentum and angular momentum of the disk, *i.e.*:

$$\begin{aligned} -(\omega/\omega_1)^2 U_{\text{out}} &= -3 [F\mathbf{e}_1 \cdot \mathbf{e}_z + QR^2/\ell^2 \mathbf{e}_2 \cdot \mathbf{e}_z], \\ -(\omega/\omega_1)^2 \Theta_{\text{out}}/2 &= -3 [F\mathbf{e}_1 \cdot \mathbf{e}_\theta + QR^2/\ell^2 \mathbf{e}_2 \cdot \mathbf{e}_\theta]. \end{aligned} \quad (11)$$

In Eqs. (11) we have introduced:

$$\omega_1 = \sqrt{\frac{ES}{\ell m}}, \quad (12)$$

together with the adimensional longitudinal force $F = \|\mathbf{F}\|\ell/(ESu_0)$, with $\mathbf{F} = ESu_0W'(1)/\ell\mathbf{e}_1$ and the adimensional shear forces $Q = -\|\mathbf{Q}\|\ell^3/(E\mathcal{I}u_0)$, with $\mathbf{Q} = -E\mathcal{I}u_0/\ell^3W'''(1)\mathbf{e}_2$. The functions F and Q are non-homogeneous polynomials of first degree in the variables U_{out} and Θ_{out} . Hence, Eqs. (11) represent a linear system for the unknown variables U_{out} and Θ_{out} . The solution of the system gives:

$$\begin{aligned} U_{\text{out}}(\omega) &= 24\omega_1^2\{r^2(\ell^2 + 12\mathcal{R}^2)\omega^2 + 18(\ell^2 - 8r^2)\mathcal{R}^2\omega_1^2 \\ &\quad + [-r^2(\ell^2 - 12\mathcal{R}^2)\omega^2 + 18\ell^2\mathcal{R}^2\omega_1^2] \cos(2\Psi)\}/\mathcal{D}(\omega), \quad \text{and} \\ \Theta_{\text{out}}(\omega) &= \left[24\sqrt{2}r^2(\ell^2 - 12\mathcal{R}^2)\omega^2\omega_1^2 \cos\Psi \sin\Psi \sqrt{(8r^2 - \ell^2 - \ell^2 \cos(2\Psi))/r^2}\right] / \mathcal{D}(\omega), \quad (13) \end{aligned}$$

with:

$$\begin{aligned} \mathcal{D}(\omega) &= 16\ell^2r^2\omega^4 + 9[\ell^4 - 96r^2\mathcal{R}^2 + 4\ell^2(\mathcal{R}^2 - 2r^2)]\omega^2\omega_1^2 - 432(\ell^2 - 8r^2)\mathcal{R}^2\omega_1^4 \\ &\quad + 3\omega_1^2\{4 \cos(2\Psi)[(\ell^4 - 2\ell^2r^2 + 24r^2\mathcal{R}^2)] + \ell^2(\ell^2 - 12\mathcal{R}^2)\omega^2 \cos(4\Psi)\}. \quad (14) \end{aligned}$$

We refer to the functions (13) as the longitudinal and torsional frf, respectively. In Fig. S8 we represent longitudinal frf $U_{\text{out}}(\omega)$ in Eqs. (13) as a function of frequency (see blue solid line) and compare it to the finite element calculation of the structure in Fig. S7. The comparison shows good agreement. The geometric and physical parameters are reported in the caption of Fig. S8. In addition, the second moment of inertia of a circular beam is $\mathcal{I} = \pi b^4/4$ which results in the gyration radius of the beams being $\mathcal{R} = \sqrt{\mathcal{I}/S} = 0.2$ mm.

Analogous conclusions hold for Fig. S9 where we show the comparison of the FE torsional frequency response function with its analytical approximation in Eq. (13). Moreover, the frf of a non-tilted TRO- ($\Psi = \pi/2$ in Eq. (13)) reduces to:

$$U_{\text{out}}(\omega)|_{\Psi=\pi/2} = [1 - 1/3(\omega/\omega_1)^2]^{-1}, \quad (15)$$

i.e. the frequency response function of a point mass m connected to the time harmonic base excitation by three parallel massless springs each of which has a linear stiffness of $\kappa = ES/\ell$.

4. **Comment:** *Reference 30 needs to be modified.*

- **Response:** We modified Reference 30 as follows: Supplementary Materials

References

- [1] J. S. Jensen. Phononic band gaps and vibrations in one- and two-dimensional mass-spring structures. *Journal of Sound and Vibration*, 266:1053–1078, 2003.

- [2] T. Delpero, G. Hannema, B. Van Damme, S. Schoenwald, A. Zemp, and A. Bergamini. Inertia amplification in phononic crystals for low frequency bandgaps. In *8 ECCOMAS SMART 2017*, pages 1–14, 2017.
- [3] A. O. Krushynska, A. Amendola, F. Bosia, C. Daraio, N. M. Pugno, and F. Fraternali. Accordion-like metamaterials with tunable ultra-wide low-frequency band gaps. *New Journal of Physics*, 20(7):073051, 2018.
- [4] A. H. Orta and C. Yilmaz. Inertial amplification induced phononic band gaps generated by a compliant axial to rotary motion conversion mechanism. *Journal of Sound and Vibration*, 439:329 – 343, 2019.
- [5] T. Frenzel, M. Kadic, and M. Wegener. Three-dimensional mechanical metamaterials with a twist. *Science*, 358(6366):1072–1074, 2017.
- [6] I. Fernandez-Corbaton, C. Rockstuhl, P. Ziemke, P. Gumbsch, A. Albiez, R. Schwaiger, T. Frenzel, M. Kadic, and M. Wegener. New twists of 3D chiral metamaterials. *Advanced Materials*, 31(26):1807742, 2019.
- [7] C. M. Soukoulis and M. Wegener. Past achievements and future challenges in the development of three-dimensional photonic metamaterials. *Nature Photonics*, 5(9):523, 2011.
- [8] J. Kaschke and M. Wegener. Optical and infrared helical metamaterials. *Nanophotonics*, 5(4):510–523, 2016.
- [9] S. Linden, C. Enkrich, M. Wegener, J. Zhou, T. Koschny, and C. M. Soukoulis. Magnetic response of metamaterials at 100 THz. *Science*, 306(5700):1351–1353, 2004.
- [10] T. Bückmann, R. Schittny, M. Thiel, M. Kadic, G. W. Milton, and M. Wegener. On three-dimensional dilational elastic metamaterials. *New Journal of Physics*, 16(3):033032, 2014.
- [11] C. Andrade, C. S. Ha, and R. Lakes. Extreme Cosserat elastic cube structure with large magnitude of negative Poisson ratio. *Journal of Mechanics of Materials and Structures*, 13(1):93–101, 2018.
- [12] R. S. Lakes. Stability of Cosserat solids: size effects, ellipticity and waves. *Journal of Mechanics of Materials and Structures*, 13(1):83–91, 2018.
- [13] A. B. Movchan and N. V. Movchan. *Mathematical modelling of solids with nonregular boundaries*, volume 3. CRC Press, 1995.
- [14] V. Kozlov, V. Maz'Ya, and A. B. Movchan. *Asymptotic analysis of fields in multi-structures*. Oxford University Press on Demand, 1999.
- [15] K. F. Graff. *Wave motion in elastic solids*. Courier Corporation, 2012.
- [16] D. Tallarico, N. V. Movchan, A. B. Movchan, and D. J. Colquitt. Tilted resonators in a triangular elastic lattice: chirality, bloch waves and negative refraction. *Journal of the Mechanics and Physics of Solids*, 103:236–256, 2017.

List of Figures

S4	Influence of the number of unit cells on the frequency response of the syndiotactic and isotactic crystals. Transmission diagram as a function of the number of unit cells (1 to 4) composing finite (a) isotactic and (b) syndiotactic crystals. The thicker red and blue curves represent the case of 2 unit cells and correspond to those reported in Figs. 5b,c of the main text. The shades of color correspond to 1, 2, 3 and 4 unit cells, respectively. Imaginary component of the wave number k as a function of frequency for the (c) isotactic and (d) syndiotactic crystal.	16
4	Influence of the number of unit cells on the twist for the isotactic and syndiotactic crystals. Surface plots of the xy component of the deformation for the (a) isotactic and (b) syndiotactic crystals (4 unit cells) subjected to an axial static load. c , The twist magnitude as a function of the number of disks for the isotactic (red lines) and syndiotactic (blue lines) crystals. The twist increases proportionally to the number of unit cells composing the structure in the isotactic crystal differently from the syndiotactic case.	17
R1	Effect of damping on the frequency response of the syndiotactic crystal. Frequency response of the syndiotactic crystal calculated for different damping coefficients. The blue, green and red curves correspond to a damping coefficient of $2.5 \cdot 1'^{-2}$, $2.5 \cdot 1'^{-3}$, and $2.5 \cdot 1'^{-6}$, respectively. It is worth noticing that for extremely weakly damped systems, the velocity amplitude ratio can indeed raise above 1 (horizontal black line).	18
S7	Schematic representation of TRO comprising three beams (dashed lines) and two rigid disks (upper and lower horizontal lines). The upper disk can rotate around and translate along the \mathbf{e}_z axis. The lower disk is constrained to translate along the \mathbf{e}_z axis and provides the base excitation to the TRO.	19
S8	TRO longitudinal frequency response function. Comparison of the FE frequency response function of a single resonator in fig. (S7) (red crosses) with its analytical approximation U_{out} in Eq. (13) (blue solid line). The parameters are $\rho = 7850 \text{ kg/m}^3$ and $E = 2.1 \times 10^{11} \text{ Pa}$. The disk has a radius $r = 2.5 \text{ cm}$ and mass $m = 0.07 \text{ kg}$. The beams are of circular cross section with radius $b = 0.5 \text{ mm}$. The length of the beams is $\ell = 3.4 \text{ cm}$ and their inclination is $\Psi = \pi/4 \text{ rad}$	20
S9	TRO torsional frequency response function. Comparison of the FE torsional frequency response function of a single resonator in Fig. (S7) (red crosses) with its analytical approximation $\Theta_{\text{out}}(\omega)$ in Eq. (13) (blue solid line). The parameters of the resonator are the same as in Fig. S8.	21

Figure S4: **Influence of the number of unit cells on the frequency response of the syndiotactic and isotactic crystals.** Transmission diagram as a function of the number of unit cells (1 to 4) composing finite (a) isotactic and (b) syndiotactic crystals. The thicker red and blue curves represent the case of 2 unit cells and correspond to those reported in Figs. 5b,c of the main text. The shades of color correspond to 1, 2, 3 and 4 unit cells, respectively. Imaginary component of the wave number k as a function of frequency for the (c) isotactic and (d) syndiotactic crystal.

Figure 4: **Influence of the number of unit cells on the twist for the isotactic and syndiotactic crystals.** Surface plots of the xy component of the deformation for the (a) isotactic and (b) syndiotactic crystals (4 unit cells) subjected to an axial static load. **c**, The twist magnitude as a function of the number of disks for the isotactic (red lines) and syndiotactic (blue lines) crystals. The twist increases proportionally to the number of unit cells composing the structure in the isotactic crystal differently from the syndiotactic case.

Figure R1: **Effect of damping on the frequency response of the syndiotactic crystal.** Frequency response of the syndiotactic crystal calculated for different damping coefficients. The blue, green and red curves correspond to a damping coefficient of $2.5 \cdot 10^{-2}$, $2.5 \cdot 10^{-3}$, and $2.5 \cdot 10^{-6}$, respectively. It is worth noticing that for extremely weakly damped systems, the velocity amplitude ratio can indeed raise above 1 (horizontal black line).

Figure S7: **Schematic representation of TRO** comprising three beams (dashed lines) and two rigid disks (upper and lower horizontal lines). The upper disk can rotate around and translate along the \mathbf{e}_z axis. The lower disk is constrained to translate along the \mathbf{e}_z axis and provides the base excitation to the TRO.

Figure S8: **TRO longitudinal frequency response function.** Comparison of the FE frequency response function of a single resonator in fig. (S7) (red crosses) with its analytical approximation U_{out} in Eq. (13) (blue solid line). The parameters are $\rho = 7850 \text{ kg/m}^3$ and $E = 2.1 \times 10^{11} \text{ Pa}$. The disk has a radius $r = 2.5 \text{ cm}$ and mass $m = 0.07 \text{ kg}$. The beams are of circular cross section with radius $b = 0.5 \text{ mm}$. The length of the beams is $\ell = 3.4 \text{ cm}$ and their inclination is $\Psi = \pi/4 \text{ rad}$.

Figure S9: **TRO torsional frequency response function.** Comparison of the FE torsional frequency response function of a single resonator in Fig. (S7) (red crosses) with its analytical approximation $\Theta_{\text{out}}(\omega)$ in Eq. (13) (blue solid line). The parameters of the resonator are the same as in Fig. S8.

Reviewers' Comments:

Reviewer #1:

Remarks to the Author:

In this thorough revision, the authors have brought forward additional analyses on several points I had raised including size effects and spatial attenuation. I appreciate the deeper investigation on the reason for the choice of the unit cell number (2 units). As per the use of the words, I would simply recommend to remove the word "believe" in this context, since the results of the additional analysis now justify their choice. Hence some rewording is needed in the new text to avoid the word believe.

I also acknowledge the key additions of several references to provide not only the right context and harmony with the previous works, but also to set the opportunity to emphasize the differences and key aspects of their work.

All the revisions including the development of a theoretical model, along with the change of the title greatly contribute to raising the overall quality and scientific level of the paper.

Reviewer #2:

Remarks to the Author:

The Authors have satisfactorily responded to all my concerns and I would suggest the present manuscript for publication.

Response to Reviewers

We thank the Reviewers for their comments and positive feedback. We are glad to learn that our revisions in response to the first round of reviews have been well received.

Reviewer #1 (Remarks to the Author):

In this thorough revision, the authors have brought forward additional analyses on several points I had raised including size effects and spatial attenuation. I appreciate the deeper investigation on the reason for the choice of the unit cell number (2 units). As per the use of the words, I would simply recommend to remove the word "believe" in this context, since the results of the additional analysis now justify their choice. Hence some rewording is needed in the new text to avoid the word believe.

In response to the observation of Reviewer #1, we have deleted the wording 'We believe that' in the section on size effect and changed the sentence to:

«In reason of this, and supported by both the numerical and experimental measurements of the transmissibility dip (see Fig. 1c of the main text), the choice of a finite structure spanning over 2 unit cells is sufficient to give a reasonable evidence of the material properties.»

I also acknowledge the key additions of several references to provide not only the right context and harmony with the previous works, but also to set the opportunity to emphasize the differences and key aspects of their work.

All the revisions including the development of a theoretical model, along with the change of the title greatly contribute to raising the overall quality and scientific level of the paper.

Reviewer #2 (Remarks to the Author):

The Authors have satisfactorily responded to all my concerns and I would suggest the present manuscript for publication.

We would like to acknowledge the contributions made by Reviewer #2, especially for encouraging us to delve deeper into the analytical modeling of the system, which has helped us gain further insights into it.

Further editorial changes have been made in response to the Editor's requests.

We look forward to next steps in the publication process.

Sincerely,

Empa

Dr. Andrea Bergamini

Senior Scientist, Acoustics/Noise Control Laboratory

andrea.bergamini@empa.ch | +41 58 765 4424

Empa

Dr. Marco Miniaci

Scientist, Acoustics/Noise Control Laboratory

marco.miniaci@empa.ch